

# Biofouling of inlet pipes affects water quality in running seawater aquaria and compromises sponge cell proliferation

Brittany E. Alexander[1,*], Benjamin Mueller[2], Mark J.A. Vermeij[2,3], Harm H.G. van der Geest[1] and Jasper M. de Goeij[1,2,*]

[1] Department of Aquatic Environmental Ecology, Institute for Biodiversity and Ecosystem Dynamics, University of Amsterdam, Amsterdam, Netherlands
[2] CARMABI Foundation, Willemstad, Curaçao
[3] Department of Aquatic Microbiology, Institute for Biodiversity and Ecosystem Dynamics, University of Amsterdam, Amsterdam, Netherlands
* These authors contributed equally to this work.

Corresponding author
Jasper M. de Goeij,
j.m.degoeij@uva.nl

## ABSTRACT

Marine organism are often kept, cultured, and experimented on in running seawater aquaria. However, surprisingly little attention is given to the nutrient composition of the water flowing through these systems, which is generally assumed to equal *in situ* conditions, but may change due to the presence of biofouling organisms. Significantly lower bacterial abundances and higher inorganic nitrogen species (nitrate, nitrite, and ammonium) were measured in aquarium water when biofouling organisms were present within a 7-year old inlet pipe feeding a tropical reef running seawater aquaria system, compared with aquarium water fed by a new, biofouling-free inlet pipe. These water quality changes are indicative of the feeding activity and waste production of the suspension- and filter-feeding communities found in the old pipe, which included sponges, bivalves, barnacles, and ascidians. To illustrate the physiological consequences of these water quality changes on a model organism kept in the aquaria system, we investigated the influence of the presence and absence of the biofouling community on the functioning of the filter-feeding sponge *Halisarca caerulea*, by determining its choanocyte (filter cell) proliferation rates. We found a 34% increase in choanocyte proliferation rates following the replacement of the inlet pipe (i.e., removal of the biofouling community). This indicates that the physiological functioning of the sponge was compromised due to suboptimal food conditions within the aquarium resulting from the presence of the biofouling organisms in the inlet pipe. This study has implications for the husbandry and performance of experiments with marine organisms in running seawater aquaria systems. Inlet pipes should be checked regularly, and replaced if necessary, in order to avoid excessive biofouling and to approach *in situ* water quality.

## INTRODUCTION

Running seawater aquaria are frequently used to study the physiology of marine organisms under controlled, *ex situ* condition (e.g., *Wilkerson & Muscatine, 1984*; *Enríquez, Méndez & Prieto, 2005*; *Anthony et al., 2008*; *Duckworth & Peterson, 2013*). In the experimental design and set-up of such studies, ambient physical abiotic factors, such as light, temperature, and water flow are given the most attention since these are well known to deviate from *in situ* conditions. However, surprisingly little attention is given to biotic and chemical abiotic factors in running seawater aquaria systems, which are usually only monitored in specific feeding or nutrient-enrichment experiments (e.g., *Tacon et al., 2002*; *Jiménez & Ribes, 2007*; *Bracken, 2004*). It is generally assumed that the chemical and biological composition of the seawater flowing through aquaria matches *in situ* ambient water. The extent to which changes in water quality occur within running seawater aquaria and the potential effect of this on the physiology of experimental marine organisms remains largely unknown.

The motivation for the present study was a large discrepancy in the number of proliferative cells measured in the sponge *Halisarca caerulea* (Porifera: Demospongiae) during two distinct fieldwork periods of several months, using the same running seawater aquaria system and identical methodology. In the first series of experiments, the proliferation rate of *H. caerulea* filter cells (choanocytes), i.e., the percentage of proliferative choanocytes after 6 h, was estimated to be $46.6 \pm 2.6\%$ (mean $\pm$ 95%-CI) under steady-state (negligible growth) conditions (*De Goeij et al., 2009*). During the second series of experiments, which were performed five to seven years later, we measured a significantly lower choanocyte proliferation rate of $17.6 \pm 3.3\%$ for the same species (*Alexander et al., 2014*; *Alexander et al., 2015*). We discussed and hypothesized the possible cause of this altered cell proliferation to be a suboptimal food supply to the aquaria during the latter fieldwork period (*Alexander et al., 2014*; *Alexander et al., 2015*). Preliminary tests also showed that during that second fieldwork period the bacterial abundances in the aquaria water were approximately three times lower ($3.0 \times 10^5$ per mL) than in water samples taken at the reef entrance of the inlet pipe ($8.8 \times 10^5$ per mL). Bacterial numbers are a good proxy indicating the food availability to sponges. The natural diet of these filter-feeding organisms mainly consists of bacterio- and phyto-plankton (e.g., *Pile et al., 1997*; *Ribes, Coma & Gili, 1999*), and dissolved organic matter (*Yahel et al., 2003*; *De Goeij et al., 2008*; *Mueller et al., 2014*). The average bacterial retention efficiency is high, ranging between 68 and 95% for a wide range of tropical-(*Mueller et al., 2014*), temperate- (*Pile, Patterson & Witman, 1996*), and cold-water (*Yahel et al., 2007*) sponge species. The low bacterial abundances observed in our running seawater aquaria could therefore indeed point toward suboptimal nutritional conditions and may explain the compromised physiology of our experimental organisms.

Biofouling is a common problem reported in power plants and desalinization factories that use running seawater (e.g., *Azis, Al-Tisan & Sasikumar, 2001*; *Railkin, 2003*), and can affect water quality and hydrodynamic patterns (*Flemming & Geesey, 1991*). The low bacterial abundances in our running seawater aquaria may, therefore, have been caused by the activity of biofouling communities, such as suspension- or filter-feeding organisms, established on the inside walls of inlet pipes. The initial cell proliferation study (*De Goeij et al., 2009*) was carried out when the aquaria inlet pipe had been in place for only a few

months, whereas the inlet pipe in the latter study (*Alexander et al., 2014*) had been in place for seven years, allowing much more time for the establishment of biofouling communities. In addition, water flowing through running seawater aquaria may experience an increase in inorganic nutrients due to excreted waste products from these biofouling communities (e.g., *Smaal & Prins, 1993*; *Southwell et al., 2008*). However, limited data is currently available on the effects of biofouling on the water quality of running seawater aquaria used to conduct physiological and ecological experiments on marine organisms.

To gain a better insight into the effect of biofouling on water quality and the outcome of experiments held in running seawater aquaria we asked the following research questions: Is water quality within our running seawater aquaria system affected by biofouling communities? If so, do bacterial abundances increase and nutrient concentrations decrease after the removal of such communities, i.e., by replacing the old inlet pipe? Does the presence of biofouling communities significantly hamper the physiology of experimental organisms kept in open seawater aquaria? In order to answer these questions, the bacterial abundances and inorganic nutrient concentrations (nitrate, nitrite, ammonium, and phosphate) were assessed in the reef water flowing along the length of the inlet pipe and in the flow-through aquarium fed by the inlet pipe. Subsequently, the presence and distribution of biofouling communities inside the 7-year old pipe was investigated. After replacing the old inlet pipe, the aforementioned water quality assessments were repeated, and the choanocyte proliferation rates for our model organism *H. caerulea* were determined before and after the installation of the new inlet pipe.

## MATERIALS AND METHODS

Fieldwork was performed under the research permit (#2012/48584) issued by the Curaçaoan Ministry of Health, Environment and Nature (GMN) to the CARMABI foundation.

### Running seawater aquaria system

The running seawater aquaria system is located on the Southern Caribbean island of Curaçao at the CARMABI research station ($12°12'$N, $68°56'$W). The land-based facility consists of 18 glass flow-through aquaria ranging in volume from 50 to 160 L. Seawater is pumped (Hayward Super Pump SP0150Z1CM; capacity $\sim$400 L min$^{-1}$) from 10 m water depth at the reef slope through a 100-m long polypropylene pipe (5 cm inner $\varnothing$). The first 60 m of the pipe lies underwater, whereas the last 40 m, including the pump, is located above-water, partially underground. Water flow is regulated separately for each flow-through aquarium. The last time the old inlet pipe of the running seawater aquaria system had been replaced was in 2006. The new inlet pipe (only the first 60 m of the underwater section) was replaced on April 6th, 2013.

### Sponge collection

All specimens of the encrusting sponge *H. caerulea* were collected from the fringing coral reefs on the leeward coast of Curaçao (Southern Caribbean, $12°12'$N, $68°56'$W) at 15–30 m water depths by SCUBA between February and March 2013, 1–2 months before the

replacement of the old pipe. Pieces of sponge were chiseled from the reef framework and the attached substrate was cleared of other organisms. All sponges were trimmed to a size of approximately 25 cm$^2$ and subsequently kept in 100-L running seawater aquaria, with a flow rate of 3 L min$^{-1}$. Aquarium water was at ambient seawater temperature (26–27 °C) and kept under natural light cycles (the semi-enclosed aquarium building receives natural daylight). Semi-transparent black plastic sheets were used to imitate *in situ* cryptic (i.e., the collected sponges inhabited coral cavity walls) light conditions that sponges experienced (photosynthetically active radiation (PAR) level 5–15 µmol photons m$^{-2}$ s$^{-1}$ during daylight hours). Prior to cell proliferation experiments, sponges acclimatized to aquarium conditions for a minimum of 1 week before and after the old pipe was replaced by the new pipe to ensure they fully recovered from the collection and transportation to the aquaria (*De Goeij et al., 2009*; *Alexander et al., 2014*; *Alexander et al., 2015*).

## Water sample collection

Water samples were taken inside the old pipe on March 22, 2013 and inside the new pipe on April 9, 2013, 3 d after its installation. To sample water from the center of the inlet pipe, Ø1.5 mm holes were drilled underwater using a hand drill through which samples were collected using 20 mL syringes with needle. The holes were sealed afterwards with marine PC-11 two-component epoxy (Protective Coating Company, Allentown, PA, USA). Triplicate 20 mL water samples were taken to determine bacterial abundances and inorganic nutrient concentrations along the length of the old and new inlet pipes at 0 (a few cm inside the inlet pipe), 3, 6, 12, 24, and 60 m from the inlet pipe entrance (Fig. 1 and Fig. S1) as well as in one of the flow-through aquaria at 100 m from the entrance of the inlet pipe.

## Analysis of bacterial abundance

Ten mL of the water sample was fixed immediately with 0.57 mL 35% formaldehyde solution (final concentration ∼2%) for 1 h at 4 °C in the dark. Fixed samples were then filtered on 0.2 µm polycarbonate filters (25 mm, Nuclepore Track-Etch; Whatman, Kent, UK) with 0.45 µm cellulose nitrate support filters (25 mm; Sartorius Stedim Biotech GmbH, Goettingen, Germany). The polycarbonate filters were air-dried and stored at −20 °C until further processing. Filters were mounted on microscope slides in DAPI-mix (final concentration 1 µg L$^{-1}$) and bacteria were counted using an epifluorescence microscope (×1,250). At least ten fields (each 0.0025 mm$^2$) of a counting grid were counted per slide, or up to a minimum of 200 bacteria, when ten fields were not sufficient.

## Analysis of inorganic nutrients

Five mL of the water sample was filtered using a 0.2 µm syringe filter (25 mm Puradisc, Whatman, UK) and stored at −20 °C in 6 mL vials (Pico Prias, PerkinElmer; Waltham, MA, USA) until further processing. The concentrations of dissolved inorganic nutrients (nitrate [$NO_3^-$], nitrite [$NO_2^-$], ammonium [$NH_4^+$], ortho-phosphate [$PO_4^{3-}$]) were measured colorimetrically on a Skalar segmented flow autoanalyzer according to the manufacturer's directions.

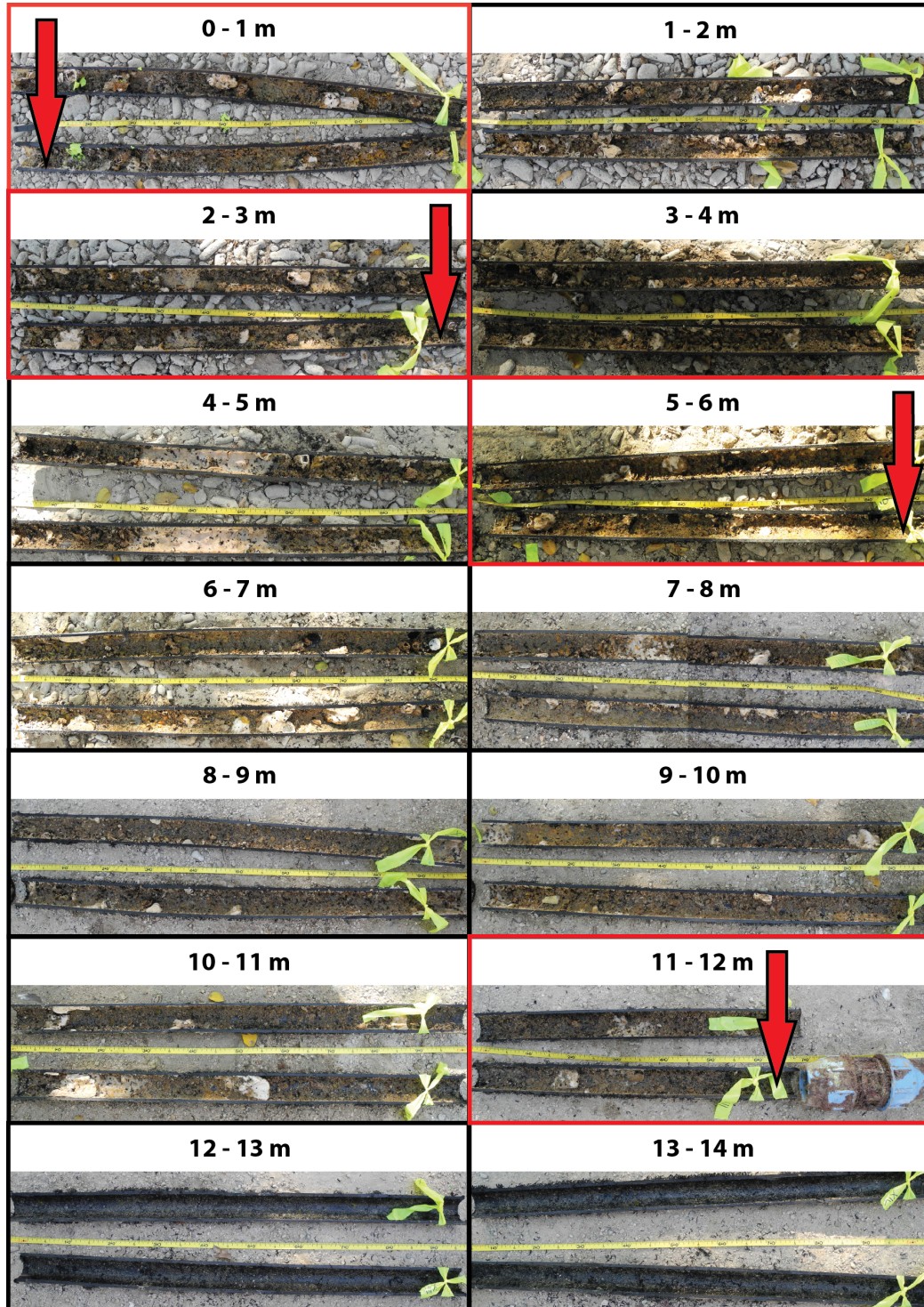

**Figure 1 Biofouling communities within the old inlet pipe.** The first 14 m from the entrance of the inlet pipe are shown. No biofouling communities were found after the initial 12 m. Water samples inside the pipe were taken from areas of the pipe outlined in red and marked by the red arrows (0, 3, 6, and 12 m).

### BrdU-labeling and sponge tissue sampling

Sponges ($n = 3$ before and $n = 3$ after installation of the new pipe; all different individuals) were enclosed in incubation chambers (3 L) with magnetic stirring devices (*De Goeij et al., 2009*; *Alexander et al., 2014*; *Alexander et al., 2015*), which were kept in the aquaria during the experiments to maintain ambient seawater temperature. In order to measure cell proliferation, 5-bromo-2′-deoxyuridine (BrdU; Sigma-Aldrich, Waltham, MA, USA) was added to incubation chambers containing the sponges. Sponges were incubated in seawater containing 50 μmol L$^{-1}$ BrdU for 6 h (continuous labeling) in order to estimate choanocyte proliferation rates (*Nowakowski, Lewin & Miller, 1989*; *De Goeij et al., 2009*; *Alexander et al., 2014*). Immediately after the incubations, one tissue sample ($\sim$0.5 cm$^2$) was taken from each sponge and fixed in 4% paraformaldehyde in phosphate-buffered saline (PFA/PBS; 4 h at 4 °C), rinsed in PBS, dehydrated through a graded series of ethanol and stored in 70% ethanol at 4 °C until further processing.

### Sponge cell proliferation

Histological sections (3 μm) of BMM-embedded sponge tissue were cut on a pyramitome (LKB 11800, UK) using glass knives and collected on glass slides (StarFrost; Knittelglass, Braunschweig, Germany). BrdU immunohistochemistry was performed according to Alexander and colleagues (*2014*; *2015*) using a mouse anti-BrdU monoclonal antibody (MUB0200S, Nordic-MUbio, Susteren, The Netherlands), which was detected with an avidin-biotin enzyme complex (Vectastain Elite ABC Kit; Vector Laboratories, Burlingame, CA, USA). BrdU-positive cells were visualized with DAB (DAKO, Glostrup, Denmark) on haematoxylin-counterstained sections, and mounted in Entellan (Merck, Kenilworth, NJ, USA). BrdU-labeled mouse intestinal tissue was used as a positive control and immunohistochemistry without primary antibody (on both mouse and sponge tissue) served as a negative control, as previously described (*Alexander et al., 2014*). All slides were examined under a light microscope (Olympus BH-2) and photographs were taken using an Olympus DP70 camera. From each tissue sample, three areas of the sponge were sectioned, each approximately 100 μm apart. At least 250 choanocytes were counted from each section making a total of at least 750 (three sections × 250 cells) cells counted per sponge.

### Analysis of biofouling in the old inlet pipe

After water samples were taken from the old pipe and tissue samples were taken from the sponges in the flow-through aquarium, the 60 m underwater section of the pipe was removed and cut into 1-m pieces over its entire length. Each 1-m piece was then cut in half along its length and photographed to assess the presence of biofouling communities (see Fig. 1 for an overview and Fig. S1 for close-up photographs of the biofouling communities in the first 12 m from the inlet pipe entrance).

### Statistical analysis

The differences in bacterial abundance and inorganic nutrients between the old and new pipe and along the length of both pipes were tested using exponential (bacteria) and linear (nutrients) models. Differences in bacterial abundances and dissolved inorganic nutrients

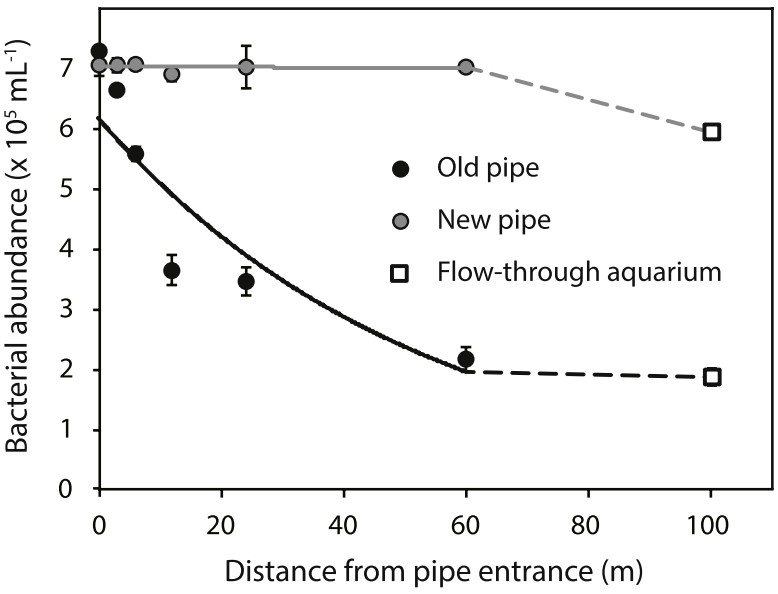

**Figure 2 Bacterial abundance along the old (black circles) and new (grey circles) inlet pipes leading to the flow-through aquarium (open squares).** Solid lines indicate exponential models for bacterial abundance within the first 60 m of inlet pipe (i.e., the section that was replaced). The dotted lines represent differences in bacterial abundance between the flow-through aquarium, which was 100 m from the inlet pipe entrance, and the replaced section of the inlet pipe.

were also tested in the water flowing through aquaria fed by the old and new pipe, using linear models. Additionally, differences in choanocyte proliferation rates between sponges kept in flow-through aquaria fed by the old and new pipe were investigated with a linear model. The significance threshold was set at 0.05. All calculations were carried out in R (see Data S1 for complete dataset (bacterial abundance and inorganic nutrients) and Supplemental Information 5 for R-scripts).

## RESULTS

### Water quality

Bacterial abundances in water at the reef entrance of the old inlet pipe ($7.3 \pm 0.5 \times 10^5$ mL$^{-1}$; mean $\pm$ SD) were 3.9 fold higher than in the flow-through aquarium ($1.9 \pm 0.1 \times 10^5$ mL$^{-1}$) (Fig. 2) of the running seawater system. Bacterial abundance decreased with increasing distance from the reef entrance of the old inlet pipe (exponential model, $p = 0.009$, Fig. 2) and the largest drop in bacterial abundance occurred in the first 12 m of the pipe. Nitrate and nitrite concentrations increased significantly with increasing distance from the entrance of the old inlet pipe (linear models, $p = 0.008$ [nitrate and nitrite], Figs. 3A and 3B). The concentration of ammonium increased from the entrance of the old inlet pipe to the flow-through aquaria, but did not significantly increase along the length of the old pipe (linear model, $p = 0.5$, Fig. 3C), and no significant difference was observed in the concentration of phosphate along the length of the old pipe (linear model, $p = 0.07$, Fig. 3D).

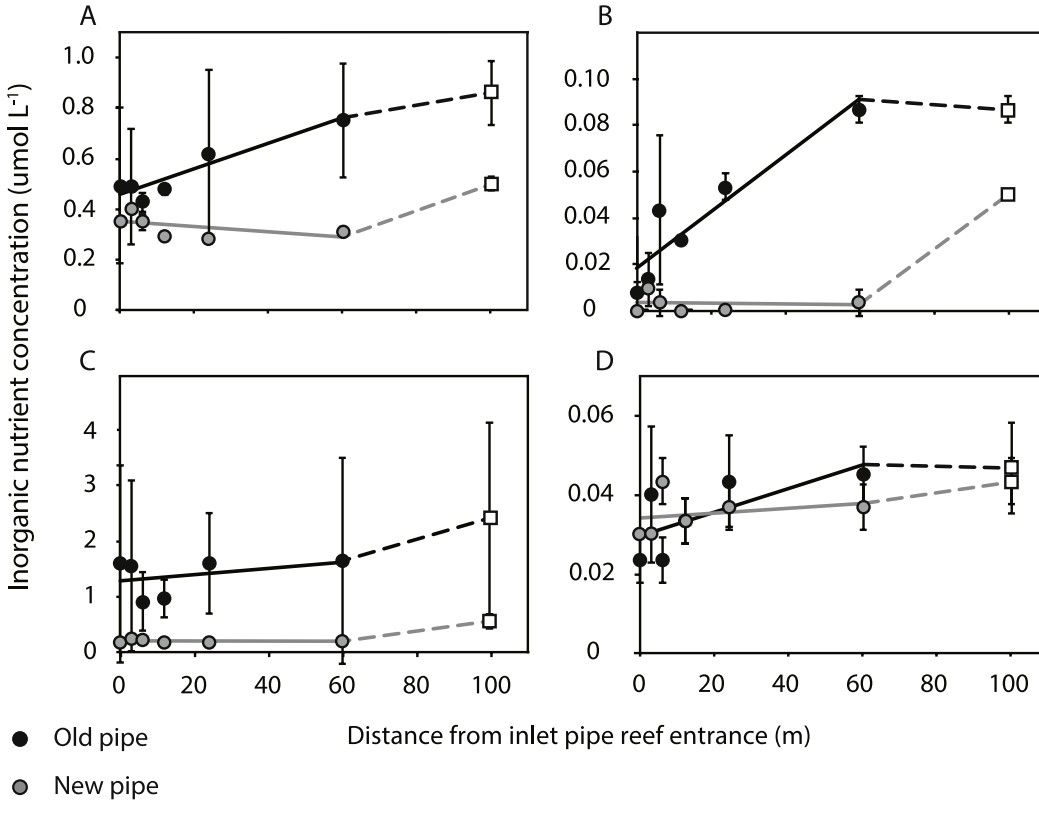

● Old pipe

● New pipe

□ Flow-through aquarium

**Figure 3** (A) Nitrate [$NO_3^-$], (B) nitrite [$NO_2^-$], (C) ammonium [$NH_4^+$], and (D) phosphate [$PO_4^{3-}$] **concentrations along the length of the old (black circles) and new pipes (grey circles), and in the aquarium (open squares).** Solid lines indicate linear models for inorganic nutrient concentrations within the first 60 m of inlet pipe (the section that was replaced), which was 100 m from the entrance of the inlet pipe. The dotted lines represent differences in inorganic nutrient concentrations between the flow-through aquarium and the replaced section of the inlet pipe.

The installation of the new pipe caused a significant change in all water quality parameters (*p* values < 0.05, Figs. 2 and 3A–3C), compared to the old pipe, except the concentration of phosphate, which remained similar (linear model, *p* = 0.77, Fig. 3D). There was no significant difference between the reef water samples (taken at 0 m from the pipe entrance) before (March 22, 2013) and after the pipe change (April 9, 2013) for any of the water quality parameters (linear model, *p* = 0.07 [bacteria], *p* = 0.12 [nitrite, ammonium, and phosphate], *p* = 0.48 [nitrate]). Bacterial abundance was significantly higher along the length of the new pipe compared to the old (exponential model, *p* = 0.03) and no longer changed along the length of the new pipe (exponential model, *p* = 0.80). However, bacterial abundance in the flow-through aquaria supplied by the new inlet pipe was lower ($5.9 \pm 0.2 \times 10^5$ mL$^{-1}$, Fig. 2A) compared to the abundance at the reef entrance of the new inlet pipe ($7.0 \pm 0.1 \times 10^5$ mL$^{-1}$). Still, bacterial abundance in the flow-through aquarium fed by the new pipe was 3.1 times higher compared to flow-through aquarium fed by the old pipe ($1.9 \pm 0.1 \times 10^5$ mL$^{-1}$). Dissolved inorganic nitrogen (nitrate, nitrite,
**Table 1 Water quality parameters (bacterial abundance [BA], nitrate ($NO_3^-$), nitrite [$NO_2^-$], ammonium [$NH_4^+$], and phosphate [$PO_4^{3-}$]) and *H. caerulea* choanocyte proliferation rates in a flow-through aquarium fed with water from the old (+ biofouling) and new inlet pipe (− biofouling).** Means ± SD are shown ($n = 3$). Percentage increases or decreases in the aforementioned parameters between the old and the new pipe are given. NA, not applicable; i.e., no significant difference.

| | Water quality parameters | | | | |
| --- | --- | --- | --- | --- | --- |
| | BA ($\times 10^5$ mL$^{-1}$) | $NO_3^-$ ($\mu$ mol L$^{-1}$) | $NO_2^-$ ($\mu$ mol L$^{-1}$) | $NH_4^+$ ($\mu$ mol L$^{-1}$) | $PO_4^{3-}$ ($\mu$ mol L$^{-1}$) |
| Aquarium water + biofouling | 1.9 ± 0.1 | 0.86 ± 0.04 | 0.09 ± 0.01 | 2.42 ± 1.73 | 0.04 ± 0.01 |
| Aquarium water − biofouling | 5.9 ± 0.0 | 0.50 ± 0.05 | 0.05 ± 0.00 | 0.57 ± 0.13 | 0.05 ± 0.01 |
| % increase or decrease from old pipe to new pipe | 217 | −42 | −42 | −77 | NA |

| | *H. caerulea* choanocyte proliferation % BrdU-positive choanocytes |
| --- | --- |
| Aquarium water + biofouling | 15.1 ± 1.9 |
| Aquarium water − biofouling | 20.2 ± 3.8 |
| % increase or decrease from old pipe to new pipe | 34 |

and ammonium) concentrations were lower in the new pipe compared to the old pipe (linear models, $p = 0.01$ [nitrite], $p = 0.01$ [nitrate], $p < 0.001$ [ammonium]) and did not change along the length of the new pipe (linear models, $p = 0.81$ [nitrite], $p = 0.81$ [nitrate], $p = 0.85$ [ammonium]). However, inorganic nitrogen concentrations were higher in the flow-through aquarium compared to the reef entrance of the new inlet pipe (Figs. 3A–3C). Replacing the old pipe, i.e., removing the biofouling community, corresponded to a 42% reduction in both nitrate and nitrite and a 77% reduction in ammonium concentrations in the flow-through aquarium (Table 1).

## Biofouling

The first 12 m of the old inlet pipe showed extensive colonization by biofouling communities (Fig. 1 and Fig. S1), which formed a cm-thick layer of living biomass decreasing its opening from Ø5 cm to approximately Ø3 cm. The abundance of biofouling organisms decreased inwards throughout the pipe and they were absent after 12 m from the reef entrance of the pipe (Fig. 1). Although an in-depth taxonomic survey of the biofouling communities was not carried out, the majority were identified as suspension- and filter-feeding organisms, including sponges, bivalves, barnacles, and ascidians.

## Sponge cell proliferation

Sponges kept in the aquaria in the presence of biofouling communities before the inlet pipe was replaced had a significantly lower choanocyte proliferation rate (15.1 ± 1.9% in 6 h) than sponges after the installation of the new inlet pipe (20.2 ± 3.8% in 6 h, linear model, $p < 0.05$, Table 1). The replacement of the pipe thus coincided with a 34% increase in the choanocyte proliferation rates of sponges kept in the aquaria over a 7-d period (Table 1).

## DISCUSSION

Here we present a study showing that biofouling of an inlet pipe feeding running seawater aquaria coincided with a decrease in bacterial abundance and an increase in dissolved inorganic nitrogen (ammonium, nitrate and nitrite) within the aquarium water. These alterations are logical consequences of the biological activity of suspension- and filter-feeding biofoulers, which included sponges, bivalves, barnacles, and ascidians (e.g., *Reiswig, 1975*; *Smaal & Prins, 1993*; *Williamson & Rees, 1994*; *Ribes et al., 2005*; *Petersen, 2007*; *Southwell et al., 2008*). Bacteria are an important food source for suspension- and filter-feeders, and the largest decrease in their abundance occurred within the first 12 m of the pipe where all biofouling organisms were found. Consequently, this created unfavorable conditions for the filter-feeding sponges kept in the running seawater aquaria system due to diminished food supplies (i.e., bacteria) and the buildup of waste products (inorganic nutrients). The new inlet pipe was free from biofouling organisms, which subsequently caused both bacterial abundance and concentrations of nitrate, nitrite, and ammonium to approach ambient reef water conditions.

Interestingly, after replacing the pipe a slight decrease in bacterial abundance and an increase in inorganic nitrogen species remained between water sampled at the entrance of the inlet pipe and the flow-through aquarium water. This could either be caused by the prevalent presence of biofouling organisms in the underground pipe section that was not replaced, or the presence of the pump in that section. The first suggestion is unlikely, since we did not find any biofouling organisms beyond the first 12 m of the pipe. Microbial biofilms may have colonized this area (*Railkin, 2003*), but these would not have caused bacterial abundances to decrease. The more likely explanation is that the pump may have caused some destruction of bacterioplankton by its impellor force (*Luckett et al., 1996*). Within the aquarist community, high pressure and cavitation caused by impellor pumps are commonly known to damage a significant proportion of the resident planktonic community (e.g., *Wijgerde, 2012* and discussed on the reef aquarist forum *Reef Central, 2008*), and could have subsequently led to a lower bacterial abundance and an increase in inorganic nutrients in the flow-through aquarium. Bacterial abundance in the aquarium fed by the new inlet pipe ($5.9 \pm 0.2 \times 10^5$ mL$^{-1}$) where nevertheless still in the range of ambient *in situ* bacterial abundances measured for Curaçaoan reef waters ($5$–$10 \times 10^5$ mL$^{-1}$) (*De Goeij & van Duyl, 2007*).

We documented a clear 34% increase in the proliferation rate of choanocytes in *H. caerulea* following the replacement of the inlet pipe. The cause of this increase cannot be unequivocally linked to changes in our water quality proxies. However, to the best of our knowledge, the only parameter that was changed in this case study was the inlet pipe. It is well established that cell proliferation is an energetically costly process, which requires a constant supply of energy and nutrients (reviewed by *Vander Heiden, Cantley & Thompson, 2009*) and can become significantly reduced, or even halted, in response to environmental stress (*Johnson & Walker, 1999*; *Jonas, 2014*). Nutrition is a pivotal determinant in the regulation of cell proliferation for the maintenance of tissue homeostasis in metazoans. Starvation causes a reduction in cell proliferation (*Chaudhary*

*et al., 2000*; *Park & Takeda, 2008*), which is reversed following re-feeding (*Aldewachi et al., 1975*). The direct relationship between nutrient concentration and cell proliferation in sponges must be investigated further, but we have found preliminary evidence that cell proliferation increases with increasing food supply, using bacterial abundance as proxy for food concentrations. Other energetically costly processes, such as reproduction or regeneration, may also cause changes in sponge cell proliferation rates. For the closely related sponge *Halisarca dujardini* it was found that up to 69.5% of their body volume could consist of reproductive elements (*Ereskovsky, 2000*) during their reproductive cycle (presumably causing less energy spent in choanocyte turnover) and cell proliferation significantly decreases during regeneration after wound infliction (*Alexander et al., 2015*). Unfortunately, to the best of our knowledge, there is no available literature on the reproductive cycle of *Halisarca caerulea*. We randomly found reproductive elements in histological sections, both oocytes and spermatic cysts, throughout both multi-months fieldwork periods, but it appeared they did not alter the choanocyte proliferation rates during experiments. During regeneration, choanocyte proliferation rates of sponge specimens residing in the aquaria fed by the old inlet pipe were reduced ($7.0 \pm 2.5\%$ in 6 h) within the first hours after wound infliction. However, after 6 days proliferation rates did not differ significantly anymore from those in steady-state tissue (i.e., the 'normal' physiological state of these sponges showing limited growth and a high turnover of choanocytes) ($12.8 \pm 1.0\%$ in 6 h) (*Alexander et al., 2015*).

Despite the 34%-increase found in this study, the proliferation rate of choanocytes for *H. caerulea* after replacement of the inlet pipe ($20.2 \pm 3.8\%$ in 6 h) was still substantially lower than found during an earlier study by De Goeij and colleagues (*2009*) ($46 \pm 2.6\%$ in 6 h). We measured cell proliferation only seven days after the old pipe was replaced and have, unfortunately, not been able to sample sponges at later time points. We hypothesize that after prolonged stress, such as malnutrition, sponges are likely to have required a longer acclimatization period to reach a higher rate of choanocyte proliferation.

## CONCLUSIONS

In conclusion, this study has implications for the husbandry of marine organisms in running seawater aquaria systems. We have found that biofouling communities residing in the inlet pipes that feed running seawater aquaria significantly alter the water quality between *in situ* ambient and *ex situ* aquarium conditions. Our results suggest that this unwillingly compromised the physiology of organisms kept in these aquaria, which can easily lead to the wrong conclusions being drawn from experimental work. This also applies to organisms that may in fact benefit from low bacterial abundances and high inorganic nutrient concentrations, such as algae and corals. Inlet pipes provide an optimal habitat for biofouling organisms due to a lack of predation, reduced competition from a lack of light, and a constant supply of food (*Polman, Verhaart & Bruijs, 2013*). They should be checked regularly—and replaced if necessary—in order to avoid excessive biofouling and to ensure that water quality in the aquaria is as close to *in situ* conditions as possible.

## ACKNOWLEDGEMENTS

We thank Dick van Oevelen and Peter van Breughel from the Royal Netherlands Institute for Sea Research, Yerseke, for their help with inorganic nutrient analysis. We thank Wim Admiraal and Ronald Osinga for commenting on drafts of the manuscript.

### Funding

This work was funded by the European Union Seventh Framework Programme (FP7/2007–2013) under grant agreement no. KBBE-2010–266033 and the Innovational Research Incentives Scheme of the Netherlands Organization for Scientific Research (NWO-VENI; 863.10.009; personal grant to JM de Goeij). The funders had no role in study design, data collection and analysis, decision to publish, or preparation of the manuscript.

### Grant Disclosures

The following grant information was disclosed by the authors:
European Union Seventh Framework Programme: KBBE-2010-266033.
Innovational Research Incentives Scheme of the Netherlands Organization for Scientific Research: NWO-VENI; 863.10.009.

### Competing Interests

The authors declare there are no competing interests.

### Author Contributions

- Brittany E. Alexander performed the experiments, analyzed the data, wrote the paper, reviewed drafts of the paper.
- Benjamin Mueller conceived and designed the experiments, performed the experiments, reviewed drafts of the paper.
- Mark J.A. Vermeij performed the experiments, reviewed drafts of the paper.
- Harm H.G. van der Geest reviewed drafts of the paper.
- Jasper M. de Goeij conceived and designed the experiments, performed the experiments, analyzed the data, contributed reagents/materials/analysis tools, wrote the paper, prepared figures and/or tables, reviewed drafts of the paper.

### Field Study Permissions

The following information was supplied relating to field study approvals (i.e., approving body and any reference numbers):

Fieldwork was performed under the research permit (#2012/48584) issued by the Curaçaoan Ministry of Health, Environment and Nature (GMN) to the CARMABI foundation.

### Data Availability

The raw data used for this manuscript is uploaded in Data S1.

## Supplemental Information

Supplemental information for this article can be found online at http://dx.doi.org/10.7717/peerj.1430#supplemental-information.

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
