# Peer review of "Biofouling of inlet pipes affects water quality in running seawater aquaria and compromises sponge cell proliferation"

_PeerJ, doi:10.7717/peerj.1430_

## Round 0.1 · original submission · Minor Revisions

Please, consider all the suggestion in the revised version of your manuscript.

Reviewer 1 ·

Basic reporting

The manuscript by Brittany et al., describes the impact of biofouling in inlet pipes of running seawater aquaria on the water quality and the potential implication for sponge cell proliferation during experiments performed in the connected aquaria. The idea of the study arose due to a large discrepancy of observed sponge cell proliferation within the same sponge species, the same seawater aquaria system and methodology, but separated by 5 to 7 years.

The conclusion of the paper is that biofouling in the seawater inlet pipes alters the water quality, i.e., bacterial cell numbers and nutrient concentrations, which potentially affects the physiology of organisms kept in the seawater aquaria - in this case the sponge species Halisarca caerulea.

Overall, the basic reporting of the study is good, but because of the more coincidentally nature of the experiment, the general conclusion drawn in this study is still somehow hampered by its limited design. See below.

Experimental design

The reason for this study is based on an accidentally finding and this what drives science forward. However, in this case this also limits the present experimental design. A replication of the experiment was not possible, since only one water inlet pipe was available.
Hence, I wonder why the authors only tested the potential effect of the renewal of the water pipe on one single sponge species (n=3)? They later state that also positive effects due to biofouling altered water quality can theoretically occur. Not testing a second sponge species (at least) or another benthic filter feeder (e.g., coral or ascidian) is a missed chance.

Moreover, not surveying the change in microbial composition genetically was surely beyond the scope of this study, but would be an interesting addition for future studies.

L 218-224 - Statistical analysis - Personally, I would like to see more background information about the used models and software. This section is to unclear in its current state.

Further questions:
Line 149-151 - What are the reasons to acclimatize the sponges at first for a minimum of 2 weeks, but then repeat the experiment after an acclimatization of just 1 week?

Validity of the findings

If the authors chose to present the significant decrease of bacterial abundance with the p value threshold of '<0.01' I wonder why they then chose '<0.05' instead of presenting the actual values. The visible bacterial and nutrient differences presented in Fig. 2 & 3 and Table 1 between the old and new pipe are convincing enough for the overall conclusion of the present experiment, even with low, but still significant p values. Personally, I like to see the real p values, especially if they are between 0.05 and 0.001.

L302-315 - The authors discuss the theoretical link between the exchange of the inlet pipe and the subsequent effect on water quality (bacterial abundance and nutrient concentration) on the sponge cell proliferation. The drawn conclusion is convincing, however, I'm missing at least one alternative explanation for the observed results. The authors suggest that further investigations are necessary and this would be a good starting point for a new working hypothesis.

Additional comments

L235 - This should be Fig. 3A, B

L296-298 - Please, elaborate how the pump could and its impellor force lead to lower bacterial abundances. Do the authors imply that the impellor force destroys microbial cells? Since, the cited paper by Luckett et al. 1996 does not specifically describe this hypothesis (as far as I know) I would appreciate a more detailed explanation at this point.

L327-329 - While the authors write in L313 to 315 that the direct relationship between nutrient concentration and cell proliferation needs further investigation, the final conclusion sounds less carefully, which is slightly contradicting. Maybe this should be rephrased in a more cautious way.

However, it is indeed very important to point out that many seawater aquarium experiments could be influenced by unrecognized effects of biofouling. Therefore, this study is overall an important contribution to its field.

Figures
I recommend to add another figure if possible. Since the biofouling of the inlet pipe is the main focus of this study, I would like to see close-up pictures of the observed biofouling within the inlet pipe in question. This could help the reader to better visualize the probably drastic change of the first 12 meters within the pipe.

The resolution of Fig. 2 is lower in comparison to Fig. 3 - please change the resolution in Fig. 2.

Reviewer 2 ·

Basic reporting

This manuscript presents experimental results of the investigation of the physiological consequences of see-water quality changes on model organisms kept in the aquaria system. In particular it was investigated the influence of the presence and absence of the biofouling community on the functioning of the filter-feeding sponge Halisarca caerulea, by determining its choanocyte proliferation rates. The authors showed, that the new inlet pipe was free from biofouling organisms, which subsequently caused both bacterial abundance and concentrations of nitrate, nitrite, and ammonium to approach ambient reef water conditions. It was also documented a clear 34% increase in the proliferation rate of choanocytes in H. caerulea following the replacement of new inlet pipe.
I think this manuscript is worth publishing in PeerJ as a Basic Reporting. The manuscript is well written: in general it is clear and the quality of the English is generally good. It is sufficiently illustrated. This paper includes new observations and thereby makes a contribution to our knowledge of the physiology and nutrition of sponges. The article will be of interest to professionals working with problems of biofouling. However, the main audience is workers using flow tanks for experiments in different biological thematic. All sections of the manuscript satisfy expected criteria. I recommend that it should be accepted for publication after amendment as itemized below.

Experimental design

In general this section contain all information relevant to repeat the study.

Validity of the findings

he data are robust, statistically sound, and controlled.

Additional comments

Manuscript n° 6646 for PeerJ entitled “Biofouling of inlet pipes affects water quality in running seawater aquaria: A case study of sponge cell proliferation“
By : Brittany E Alexander, Benjamin Mueller, Mark JA Vermeij, Harm G van der Geest, Jasper M de Goeij


This manuscript presents experimental results of the investigation of the physiological consequences of see-water quality changes on model organisms kept in the aquaria system. In particular it was investigated the influence of the presence and absence of the biofouling community on the functioning of the filter-feeding sponge Halisarca caerulea, by determining its choanocyte proliferation rates. The authors showed, that the new inlet pipe was free from biofouling organisms, which subsequently caused both bacterial abundance and concentrations of nitrate, nitrite, and ammonium to approach ambient reef water conditions. It was also documented a clear 34% increase in the proliferation rate of choanocytes in H. caerulea following the replacement of new inlet pipe.
I think this manuscript is worth publishing in PeerJ as a Basic Reporting. The manuscript is well written: in general it is clear and the quality of the English is generally good. It is sufficiently illustrated. This paper includes new observations and thereby makes a contribution to our knowledge of the physiology and nutrition of sponges. The article will be of interest to professionals working with problems of biofouling. However, the main audience is workers using flow tanks for experiments in different biological thematic. All sections of the manuscript satisfy expected criteria. I recommend that it should be accepted for publication after amendment as itemized below.

Specific comments:
Introduction
The introduction contains the pertinent literature on the subject.

Lines 76-78. You write: “We hypothesized the possible cause of this altered cell proliferation to be a suboptimal food supply to the aquaria during the latter fieldwork period”. My question is, if you tested a correlation of this altered cell proliferation with different stages of reproductive cycle in H. caerulea? Do you know the reproductive cycle and reproductive effort of your model sponge? For example, in Halisarca dujardini during their reproduction the reproductive effort (total volume of reproductive element / volume of parental mesohyl) could be up to 70% (Ereskovsky, 2000).

Materials and Methods
In general this section contain all information relevant to repeat the study.
Sponge collection section
It is not clear did you use the pieces from the same sponge individuals for experiments with old and new pipes? Or there were from different individuals?
Water sample collection section
Did you analyze water samples in situ i.e. from 10 m water depth at the reef slope on March 22, 2013 and on April 9, 2013? If it was any differences of microbes and inorganic components concentration between these dates?


Results
Line 230: Change Fig. 2A,B with Fig. 3 A,B.

---

## Round 0.2 · accepted · Accept

The revisions were appropriate. Thank you for giving us the possibility of considering your work.